# Adipose Tissue: A Novel Target of the Incretin Axis? A Paradigm Shift in Obesity-Linked Insulin Resistance

**DOI:** 10.3390/ijms25168650

**Published:** 2024-08-08

**Authors:** Michelantonio De Fano, Massimo Malara, Cristiana Vermigli, Giuseppe Murdolo

**Affiliations:** Complex Structure of Endocrinology and Metabolism, Department of Medicine, Azienda Ospedaliera Santa Maria Misericordia, Ospedale di Perugia, 06081 Perugia, Italy; massimo.malara9@gmail.com (M.M.); cri.vermigli@gmail.com (C.V.); gmurdolo@tiscali.it (G.M.)

**Keywords:** adipose tissue (AT), glucagon-like peptide-1 (GLP-1), glucose-dependent insulinotropic peptide (GIP), type 2 diabetes mellitus (T2D), obesity, inflammation

## Abstract

Adipose tissue (AT) represents a plastic organ that can undergo significant remodeling in response to metabolic demands. With its numerous checkpoints, the incretin system seems to play a significant role in controlling glucose homeostasis and energy balance. The importance of the incretin hormones, namely the glucagon-like peptide-1 (GLP-1) and the glucose-dependent insulinotropic peptide (GIP), in controlling the function of adipose cells has been brought to light by recent studies. Notably, a “paradigm shift” in reevaluating the role of the incretin system in AT as a potential target to treat obesity-linked metabolic disorders resulted from the demonstration that a disruption of the GIP and GLP-1 signaling axis in fat is associated with adiposity-induced insulin-resistance (IR) and/or type 2 diabetes mellitus (T2D). We will briefly discuss the (patho)physiological functions of GLP-1 and GIP signaling in AT in this review, emphasizing their potential impacts on lipid storage, adipogenesis, glucose metabolism and inflammation. We will also address the conundrum with the perturbation of the incretin axis in white or brown fat tissue and the emergence of metabolic disorders. In order to reduce or avoid adiposity-related metabolic complications, we will finally go over a potential scientific rationale for suggesting AT as a novel target for GLP-1 and GIP receptor agonists and co-agonists.

## 1. The Remodeling of Adipose Tissue

Adipose tissue (AT) is considered as a remarkably active and plastic organ with functional pleiotropism and elevated remodeling capacity [1,2]. In fact, AT is remarkably capable of expanding to more than double its original size by hypertrophy (enlargement of the cell size) and/or hyperplasia (increase in cell number) [3]. From a physio-pathological perspective, cellular remodeling can either lead to allostatic overload with unfavorable perturbation of both metabolic and cardiovascular control, or it can arrange nutritional adaptations in answer to variations of energy balance to preserve biological stability (i.e., homeostasis).

In conditions of over-nutrition, AT expands by increasing triacylglycerol (TAG) storage in adipocytes and goes through dynamic, metabolic and cellular adaptive changes that preserve other organs from lipotoxicity [2]. The hypertrophy of preexisting adipose cells, genesis of new adipocytes from undifferentiated precursor cells (i.e., adipogenesis), extracellular matrix proteolysis and expansion of the tissue vascular network (i.e., angiogenesis) constitute the principal mechanisms through which AT remodeling in obesity can turn into a morbid process. Furthermore, intricate interactions between various resident cell types within the expanded fat happen through cell-to-cell contact or autocrine-paracrine pathways. Non-immune and progenitor cells play a crucial role in these interactions, which contribute to the endocrine and metabolic dysregulation of unhealthy fat [2].

Even with this extraordinary adaptability, the aptitude of AT to handle continuous energy excess is limited. The limit of fat mass expansion appears to be genetically and/or environmentally determined for any different subject. Additionally, it has been suggested that the inhibition of new adipose cell formation is a trigger for the development of insulin resistance (IR), principally in the subcutaneous (sc) fat depots (i.e., abdominal and tight fat). This provocative concept (“expandability hypothesis”) justifies in part the seeming paradox of an increased risk of IR under circumstances when body fat is both reducing (i.e., lipodystrophy) and expanding (i.e., adiposity) [4,5].

It is theoretically possible for “healthy”, “unhealthy” or pathological processes to cause AT accrual [6]. Essentially, although the “appropriate” expansion of AT requires a deeply coordinated response from a variety of resident cell types, such as immune cells, endothelial precursor cells, and (pre)adipocytes, this metabolic process is mainly brought about by combination of adipocyte hypertrophy of preexisting cells and hyperplasia (i.e., the recruitment and differentiation of new adipose precursor cells) [7]. These two patterns appear tightly entwined: the link between obesity and adipose cell size is undoubtedly curvilinear, and as numerous elegant investigations have been demonstrated in studies [8,9,10], the formation of new adipocytes is triggered when the increase in adipocyte size arrives at a plateau [7]. It has been shown that nonobese subjects with type 2 diabetes mellitus (T2D) and nondiabetic subjects with a genetic predisposition for T2D (first-degree relatives: FDR) have an unwarranted hypertrophic expansion of subcutaneous adipose tissue (SAT) [10]. Because of limited adipogenesis and fat organ dysfunction, adipocyte hypertrophy may constitute a premature, obesity-independent, marker of IR and future risk of T2D. The incapacity to engage new adipose cells “on demand” to store excess lipids in the SAT leads to the hypertrophic (unhealthy) growth of AT, which is further characterized by the infiltration of macrophages and lymphocytes, extensive fibrosis, limited angiogenesis and ensuing hypoxia, all of which define the morbid fat mass accrual.

On the other hand, a rise in AT regarded as “healthy” and protective results from the effective recruitment of precursor cells to adipogenesis, together with appropriate angiogenic response, the remodeling of the extracellular matrix (ECM), and minimal inflammation [2]. Nonetheless, over time, these healthy or unhealthy fat patterns may change in response to pathophysiological or therapeutic signals, highlighting the relevance of AT plasticity in preserving cardio-metabolic balance [2].

## 2. The Incretin Hormones

The hormones glucagon-like peptide-1 (GLP-1) and glucose-dependent insulinotropic polypeptide (GIP) are secreted from different cells located in the gut, respectively, L cells (whose concentration increases from the lower jejunum to the ileum and then further to the colon and rectum) and K cells (primarily present in the duodenum and upper jejunum) [11,12]. As in animals, the primary stimulus for the secretion of these gut hormones is the absorption of nutrients, such as glucose, other carbohydrates, proteins and TAG, through the enterocytes.

This is on basis of the “incretin effect”, defined as a two-to-three-fold higher insulin secretory response to oral as compared to intravenous glucose administration, in a glucose-dependent manner and due to the direct action of GLP-1 and GIP on beta-cells [13]. Additionally, due to their insulinotropic action, incretin hormones regulate glucagon release: GIP has been found to stimulate glucagon secretion, particularly at lower glucose values [14], whereas GLP-1 suppresses glucagon release and, consequently, reduces hepatic glucose production [15].

It is well-known that the incretin effect is weakened or no longer present in subjects with T2D, albeit more or less normal secretion of GIP e GLP-1 [16,17,18]. Nevertheless, whereas the insulinotropic effects of GLP-1 are only marginally impaired in T2D, GIP has lost much of its action for largely unknown reasons [19]. Thus, GLP-1 has become a fundamental compound of incretin-based glucose-lowering medications, constituted for many years by inhibitors of dipeptidyl peptidase-4 (DDP-4i) and, specifically, GLP-1 receptor agonists (GLP-1 RAs). The latter have shown a significant impact on body weight [20], appetite [21], gastric emptying [22] and, moreover, some of these (liraglutide, dulaglutide, semaglutide sc) have shown a crucial cardiovascular protective effect in specific Cardiovascular Outcome Trials (CVOTs), both in subjects with T2D [23] and obesity [24].

The recent finding that GIP/GLP-1 receptor co-agonists like tirzepatide have higher efficacy as compared with selective GLP-1 RAs as regards glycemic control and body weight, and the consequent approval for the treatment of T2D and obesity, reinvigorated interest in GIP, which in the past was thought to be without any therapeutic potential [25].

## 3. Expression of GLP-1 and GIP Receptors in Animal and Human Tissues

GLP-1 receptors (GLP-1R) and GIP receptors (GIP-R) are expressed in many tissues in animals and humans [14]. It is thus arguable that the actions of these incretins are pleiotropic and tissue-specific (Table 1).

As expected, GLP-1R are abundantly expressed on pancreatic beta-cells but only in a minor subpopulation of alfa-cells (10–15%). Instead, GIP-R are broadly expressed both on beta and alfa cells. Their presence is also found on delta and PP cells, whereas the presence of GLP-1R on these cell populations is uncertain ([14] and references therein, [26]).

Hepatocytes and muscle cells are devoid of GLP-1R and GIP-R but some studies, that will be explored later in the manuscript, have shown how the incretin hormones can impact hepatic and muscle metabolism via indirect signaling. Nevertheless, GLP-1R may be present at low levels on hepatic stellate and/or Kupffer cells [26].

Both receptors are expressed in the heart and in blood vessels, and these findings are relevant in consideration of the cardiovascular benefits reported with some GLP-1 RAs (CVOT for tirzepatide is ongoing) [27].

In the human kidney, GLP-1R are located in preglomerular vascular smooth muscle cells, hilar and intralobular arteries and in juxtaglomerular cells, promoting, among other effects, a natriuretic action by GLP-1. On the other hand, no prominent direct effect has been seen hitherto for GIP [14,28].

GLP-1R and GIP-R have been identified in brain regions involved in the regulation of appetite, satiety, food/energy intake and energy expenditure, which is consistent with the significant impact on body weight observed with incretin therapy. These areas are primarily found in the brain stem (area postrema, nucleus tractus solitarii, lateral parabrachial nucleus), as well as the hypothalamus (arcuate, paraventricular, ventromedial and dorsomedial nuclei) [29,30]. Furthermore, the action of GLP-1 and GLP-1 RAs (especially short acting) on gastric and intestinal cells causes, respectively, the deceleration of gastric emptying [31] and the reduction in intestine motility [32], which both impact the absorption of nutrients. Instead, the impact of GIP on the gastrointestinal tract appears less decisive.

GIP-1R and/or GIP-R in other brain regions (such as the substantia nigra, hippocampus and amygdala) may be also implicated in anti-apoptotic effects, synaptic plasticity, memory, reward functions and emotional responses [33], enough to be studied in specific ongoing trials regarding neuro-degenerative disorders.

It has been reported that GLP-1R is present in the testis [34], and it has been shown that GLP-1 RAs can lower testicular inflammation and improve the quality of sperm in obese mice [35]. Furthermore, GLP-1R may also be found in the corpus cavernosum, supporting the benefits of GLP-1 RAs for erectile function [36].

Finally, GIP-R have been found in bone tissue and their activation stimulates osteoclast and osteoblasts implicated in bone remodeling [14].

## 4. Molecular Mechanisms of GLP-1 and GIP

The molecular mechanisms underlying the effects of incretin hormones at the cellular level are complex, cell-specific, and not fully understood. We will thus briefly outline the main signaling underlying the effects of GLP1 and GIP on glucose homeostasis.

Insulin release from pancreatic beta cells is, by far, the most well studied mechanism of intracellular signaling by GLP-1R ([37], and references therein).

In this system, GLP-1R carries out its “canonical” functions by binding to a G-protein coupled receptor (GLP-1R) expressed on the surface of many cells including beta cells. Upon stimulation, this leads to adenylate cyclase’s rapid activation, which raises levels of cyclic AMP (cAMP). cAMP directly activates protein kinase A (PKA) and cAMP-regulated guanine nucleotide exchange factor 2 (EPAC2), that in concert act to produce downstream signals resulting ultimately in increased insulin secretion. Mechanisms include ATP-sensitive K^+^ channel closure, the facilitation of calcium channels (VGCCs) opening, inhibiting membrane repolarization via K^+^ channels and Ca^2+^-induced Ca^2+^ release from cytoplasmic storages sites.

Through extracellular signal-regulated kinases (ERK-1 and -2), GLP-1R signaling triggers the recruitment and signaling of β-arrestin. Nevertheless, non-Ga_s_-dependent processes have also been postulated, in addition to the recruitment of the canonical Ga_s_/cAMP/PKA-dependent signal pathway. A Gα_q_-dependent pathway is suggested by experimental evidence in HEK cells through the activation of protein kinase C (PKC) and ERK-1 and -2. Under these circumstances, Gα_q_ activation triggers a PLC-mediated increase in inositol triphosphate (IP3), which raises intracellular calcium levels by releasing calcium storages from the endoplasmic reticulum calcium. PKC is stimulated by rising calcium concentrations, and PKC phosphorylates ERK-1 and -2, which in turn phosphorylates GLP-1R’s C-terminus. The internalization is then brought on by this C-terminal phosphorylation via an unclear mechanism.

Intriguingly, GLP-1R activation may differ in line with spatial and temporal factors. Receptor compartmentalization may limit the action of downstream signaling partners [38]. The generation of microdomains has been reported to influence GLP1R signaling: the PKA-induced activation of ERK-1 and -2 seems to be momentary and causes its translocation to the nucleus, while GLP-1R β-arrestin-induced ERK 1/2 activation, rather than inducing translocation, preferentially acts on cytoplasmic targets [39]. Transiently, the action of GLP-1R is regulated via the timing of cAMP fluctuations in beta cells, leading to prolonged rises of cAMP triggering a nuclear translocation of activated PKA [40].

As far as GIP is concerned, after binding its receptor on beta cells, GIP transduces its biological effects by increasing in cAMP and intracytoplasmic Ca^2+^ levels, in addition to the activation of PI3K, AKT, PKA, p38 MAP kinases and phospholipase A2 [41]. By stimulating the production of cAMP, GIP induces a positive regulation of ERK-1 and -2 via PKA and Rap1 pathways. Studies performed in cultured cells have shown that GIP can influence the activity of all known kinases of the ERK-1 and -2 module, including Mek1/2, Raf-1 and p90 RSK [42]. The main result is an increase in the size of the insulin granule pools that are immediately available for release due to the acceleration of granule mobilization. Moreover, GIP up-regulates the transcription and biosynthesis of the beta cell insulin gene as well as the expression of the glucose sensor components in beta cells.

Recently, a gut–brain axis has been discovered that links the GIP effect on hypothalamic metabolic signaling to obesity-related leptin resistance. In fact, it has been suggested that, in obesity, elevated circulating GIP levels lead to the activation of EPAC/Raf1 signaling and this is an important condition for inducing neural leptin resistance [43].

## 5. GLP-1 and GIP Actions in Liver and Skeletal Muscle

Some studies reported the effects of GLP-1 and GIP on glucose and lipid metabolism in the liver, despite the apparent lack of their specific receptors in these tissues.

Human hepatic glucose production is reduced by 20–30% when GLP-1 is infused during the experimental setting with clamped insulin and glucagon concentrations [44]. Numerous bodies of evidence suggest that GLP-1 effects on hepatic glucose metabolism may be neurally mediated, even though the exact processes behind this result are as yet unknown ([45]; and references therein). In actuality, GLP-1 administered through the central nervous system and specifically into the arcuate nucleus of the hypothalamus increases the production of glycogen and improves insulin sensitivity in the liver. Furthermore, it has been shown that there are neuronal GLP-1 sensors in the hepatic portal venous system’s vasculature, where GLP-1 signaling affects fasting and post-prandial glycemia.

Furthermore, GLP-1 has also been suggested to affect other aspects of hepatic metabolism. As a matter of fact, GLP-1 promotes fatty acid oxidation in the liver by transcriptional effects on key enzymes controlling beta oxidation [46,47]. Moreover, in obese mice, the administration of GLP-1 agonists reduced hepatic steatosis [48], an effect that has also been replicated in humans [47,49]. This important finding has paved the way to the interesting line of research about the use of incretin-based drugs for the treatment of metabolic dysfunction-associated steatotic liver disease (MASLD), which represents one of the most important morbidities in subjects with T2D and/or unhealthy obesity.

There is some debate over GIP’s impact on the liver. Research conducted on animal models indicates that GIP enhances lipid deposition in the liver, and that this process can be stopped by genetically induced GIP signal suppression. GIPR-/- mice exhibit considerably lower expression of the inflammatory adipokine IL-6, which is regulated by the suppression of inflammatory cytokine signaling (SOCS3), indicating that GIP may play a role in inflammatory change within a fatty liver ([50]; and references therein).

In humans, the pathophysiology of metabolic dysfunction-associated steatohepatitis (MASH), that constitutes an advanced stage of MASLD has also been linked to the increased post-prandial release of GIP. Specially, individuals with MASH show a prolonged GIP rise following the ingestion of a high fat load, with respect to age, body mass index, and sex-matched healthy controls. Nonetheless, there seems to be a clear correlation between the rate of hepatic steatosis and the GIP response [51].

Literature provides some evidence that high-glycemic-index (GI) food consumption is associated with excessive liver fat storage, which may be mediated by GIP. In this regard, it has been shown that acarbose, an α-glucosidase inhibitor, lowers GIP release and delays the absorption of glucose, which results in a decrease in the amount of fat in the liver [52,53].

As far as skeletal muscle is concerned, the data on GLP-1 and GIP effect are still scant. Animal studies suggest that GLP-1 enhances glycogen synthesis and increases glucose uptake in skeletal muscle. Furthermore, GLP-1 and exendin-4 (a GLP-1 R agonist) stimulate glycogen synthase activity and glucose metabolism in rat soleus muscle and human skeletal muscle [54]. Recently, in a murine model, it has been shown that GLP-1 overexpression, in skeletal muscle, causes an increase in glycogen synthesis and induces the translocation of GLUT4 to the sarcolemma, generating augmented glucose uptake. Furthermore, GLP-1 overexpression induces a myofiber-type switch towards oxidative, high-endurance muscle fibers, and also a boost of mitochondrial content and oxidative phosphorylation. All things considered, these results point to the possibility that GLP-1 increases physical endurance by causing skeletal muscle remodeling [55].

The effects of GIP on skeletal muscle are also poorly explored. O’Harte et al. showed that GIP enhanced glucose transport in the mouse diaphragm [56]. Also, GIP seems to promote skeletal muscle’s absorption of glucose in murine experimental models, both in the resting state and following insulin stimulation, in a PI3K-dependent manner [57]. However, despite these interesting observations, the mechanisms underlying these actions remain basically unclear.

## 6. Expression and Activity of GLP-1 Receptors in Adipose Tissue

### 6.1. White Adipose Tissue and GLP-1R Expression in Adipocytes and Stroma Vascular Fraction: A Depot-Dependent Difference?

Since the 1990s, reports have indicated that isolated mouse and human adipocytes contain GLP1-R ([58], and references therein). According to experimental research, GLP-1 can certainly cause both lipogenic and lipolytic effects in a dose-dependent manner, with lipogenic action predominating at picomolar concentrations and lipolytic action at nanomolar doses [59,60,61,62,63].

The presence of GLP-1R and its role have been extensively examined in murine 3T3-L1 adipocytes. In these cell-lines, GLP-1 promotes pre-adipocyte differentiation and the inhibition of apoptosis. Furthermore, GLP-1 may increase insulin-dependent glucose uptake by mechanisms already mentioned [63,64]. Accordingly, GLP-1 and exendin-4 treatment enhanced the differentiation of murine 3T3-L1 adipocyte precursors [64]. This enhanced intracellular lipid storage and insulin-mediated glucose uptake promoting intracellular lipid storage and insulin-mediated glucose uptake via the up-regulation of GLUT4 expression and phosphorylation of the insulin receptor in adipose cells [65].

It has been supposed that the promotion of adipogenesis through the activation of GLP-1R enhances the capacity of visceral hypertrophic adipocytes to store lipids, thus decreasing the ectopic lipid store and improving IR, as demonstrated by Hoffstedt et al. [66]. As previously mentioned, GLP-1 may stimulate lipolysis in adipocytes involving downstream adenylate cyclase/cAMP signaling. This “hormesis-like” response may be implicated in the different effects on weight loss observed in subjects when treated with GLP-1 RAs. Besides reestablishing “physiological” levels in the bloodstream, GLP-1 RA treatment is often related with “supraphysiological” concentrations with anorexigenic and lipolytic effects that, ultimately, contribute to weight loss.

Vendrell et al. [58] firstly reported that GLP-1R in human AT is likely correlated with the degree of IR and that there is a definite up-regulation of GLP-1R at the gene and protein levels in the visceral fat depot in individuals with morbid obesity (OB) and a high degree of IR. It is interesting that these authors showed that mature adipocytes and stromal vascular fraction (SVF), which include mesenchymal stem cells (MSCs), preadipocytes, fibroblasts, vascular endothelial cells and a variety of immune cells like AT macrophages, expressed GLP-1R at both the mRNA and protein levels. Furthermore, immunohistochemical analysis unveiled the existence of GLP-1R in mesothelium, SVF and adipose cells [58]. The picture displayed by GLP1-R expression in the study cohort showed different behavior, depending on the depot analyzed. In SAT, although OB subjects exhibited a substantial increase in GLP-1R expression when compared with non-morbidly obese counterparts (i.e., lean, overweight and obese), similar subcutaneous adipose GLP-1R expression was seen when morbidly obese subjects with a different degree of IR (i.e., low vs. high degrees of IR) were compared. On the contrary, the functional classification of OB individuals based on the degree of IR revealed that, in the context of a very high degree of IR, GLP-1R in VAT depots was significantly up-regulated. It is reasonable to hypothesize that, in the presence of extreme IR, the up-regulation of GLP1-R expression in the visceral fat depot may represent a potential compensatory mechanism to overcome commensurately lower circulating GLP-1 levels based on the discrepancies observed in GLP-1R expression in VAT depots from OB-IR patients. Accordingly, numerous studies have shown that following glucose overload, IR and diabetes clearly exhibit a decrease in circulating GLP-1 [58,67].

GLP-1R expression in VAT has unquestionably been linked to the degree of IR and with the greatest improvement in insulin sensitivity following biliopancreatic diversion surgery, even though GLP-1R expression in the SAT depot did not appear to differ considerably according to different degrees of IR [68]. This relationship is maintained by recent data in subjects with T2D, but it also reveals a negative correlation between insulin levels and GLP-1R expression in VAT. This is consistent with the findings that weight, waist circumference, blood pressure and GLP-1R expression are negatively correlated in this fat depot. In other words, it is possible to speculate that greater metabolic disturbances are associated with (and partly predicted by) the lower GLP-1R expression in the visceral fat depot. However, these data are pertinent only to individuals with morbid obesity and T2D, and it is not possible to generalize these results to other settings (i.e., non-morbidly obese or overweight subjects with insulin resistance). On the other hand, these results prompt the possibility that obese “metabolic unhealthy” subjects may exhibit lower disrupted GLP-1/GLP-1R signaling in fat tissue, which would lower local GLP-1 sensitivity. Evaluating the predictive value of pre-surgery GLP-1R expression in AT on glucose metabolism and weight loss after various bariatric procedures, no link was found between GLP-1R expression in AT and either diabetes remission, IR improvement or weight loss after surgery. Moreover, GLP-1R expression in AT is not very useful in predicting incretin response in people with morbid obesity and T2D. It also does not seem to have an impact on the incidence of certain metabolic outcomes like improved insulin sensitivity, T2D remission, and weight loss following bariatric surgery. More investigations are necessary to better examine the potential paracrine action of this receptor in AT [68].

The epicardial AT, the heart’s counterpart in visceral fat, was found to express GLP-1R [69]. Surprisingly, an excess of epicardial fat has been linked to IR and cardiovascular disorders, significantly influencing atherosclerosis through the paracrine release of pro-inflammatory cytokines [70]. It is commonly known that an increase in abdominal VAT is associated with a raised risk of cardio-metabolic issues and an increased ability to predict death [71,72]. Inflammatory mechanisms resulting from AT are partially responsible for the majority of these harmful events. In fact, new research suggests that GLP-1 RAs may reduce the production of pro-inflammatory cytokines and immune cell infiltration to reduce inflammation in many tissues, including AT [73].

Numerous GLP1-RAs demonstrated anti-inflammatory properties. For instance, exendin-4 prevents macrophages from expressing TNF-α in response to LPS [74]. Furthermore, GLP-1 RAs prevent resident macrophages in AT from being activated by nuclear factor-kappa B (NF-κB) [73]. Moreover, GLP-1 RAs have an anti-inflammatory effect on endothelial vascular cells through a number of molecular mechanisms, including the activation of adenosine monophosphate-activated protein kinase (AMPK) [75]. As much as GLP-1R expression in whole AT has been described either for mature adipocytes or the cellular constituents of SVF, it is worth understanding at this juncture which cell-type is the predominant target of the anti-inflammatory action of GLP-1.

### 6.2. Sex Differences in Response to GLP-1 RA Therapy in Type 2 Diabetes and Obesity: Prefential Effects in Subcutaneous vs. Visceral Fat Depot

Obesity and T2D are characterized by sexual dimorphism, which appears to be an important determinant of presentation, diagnosis, disease progression and complications, and, therefore, may be important for individual treatment approaches ([76,77,78]; and references therein).

As far as T2D is concerned, it has been reported that, in men, medical diagnosis occurs at an earlier age and at a lower BMI than in women, whereas females with diabetes exhibit a larger degree of obesity compared to males, despite the overall higher prevalence of overweight/obesity in the latter than in the former sex [79]. Yet, differences in adipose tissue distribution may well explain such a difference since abdominal obesity, a hallmark of the metabolically unhealthy obese phenotype, is present in almost 70% of women diagnosed with T2D as compared to approximately 40% of men [80]. On the other hand, females with T2D exhibit a worse prognosis than men showing higher cardiovascular disease (CVD) risk and all-cause and CVD-related mortality [79,81].

Although the pathophysiology underlying these sex-specific differences is beyond the scope of the present manuscript, estrogen and/or testosterone are strongly implicated in the regulation of both fat mass distribution and glucose homeostasis. In this scenario, since the pleiotropic actions of GLP-1 RAs involve tissues and processes that are hormonally regulated, it is of great interest to ask whether a sexual dimorphism may characterize the response to these agents in the treatment of obesity and/or T2D.

Regarding the efficacy of glycemic outcomes, sex-specific differences have previously been reported in relation to other antidiabetic drugs. For example, studies in the literature showed that men treated with insulin glargine or dapagliflozin, a sodium glucose cotransporter 2 inhibitor (SGLT2-i), were more likely to achieve better glycemic target in comparison to women [82,83]. On the contrary, metformin appears to exhibit similar glycemic efficacy in both sexes [84].

#### 6.2.1. Glucose Control

Currently, the literature provides conflicting evidence about a sexual dimorphism on the anti-hyperglicemic effect of GLP-1 RAs in subjects with T2D. In a retrospective pool analysis of subjects with T2D receiving exenatide (twice daily) or dulaglutide (once weekly), a reduction in HbA1c occurred irrespective of sex [85]. On the other hand, other studies highlighted a sex-specific response to GLP-1 Ras, indicating that females exhibit a wider reduction in HbA1c levels when compared with males after treatment with exenatide plus metformin [86]. Additional benefits of this combined treatment were more clearly defined in females with regard to inflammation, beta cell function and adiponectin levels (used as an indicator of insulin sensitivity), supporting the notion that sexual dimorphism is a response to the combined regimen. Analogous conclusions were also reported in another analysis examining the efficacy of liraglutide, which showed that, when stratified by age, women showed a greater reduction in HbA1c values in the 18–64 age group, whereas in the over 65 age group, predominance in men was reported [87]. It is thus arguable that these discrepancies could be explained by age-dependent hormonal variations, mainly linked to the decrease in estrogen levels in females after menopause. Lastly, further research suggested a superior response in males to treatment with GLP-1 RAs [88].

In summary, overall observations (including data from clinical trials) did not allow firm conclusions to be drawn about a possible sex-specific effect of GLP-1 RAs on glucose control.

#### 6.2.2. Weight Loss

Weight loss represents one of the most important benefits of GLP-1 RA treatment. Although the different GLP-1 RAs do not allow the same results to be achieved in terms of weight loss, cumulative evidence suggests that, even by using different representatives of this class, weight reduction is similarly more pronounced in females than in males ([76]; and references therein). The mechanism behind such a sexual dimorphism remains basically unclear. However, it has been suggested that this effect could partly be due to increased drug exposure observed in females. The results of pharmacokinetic analysis revealed that women had a 32% higher liraglutide exposure than men did at similar weights, establishing the female sex as an independent predictor of successful weight loss [89]. On the other hand, caution is required in generalizing this apparent relationship between sex-specific difference in weight loss and dissimilar drug exposure to the clinical realm. Worthy to note, it has been shown that during treatment with GLP-1 RAs, subjects with higher percentages of gastro-intestinal (GI) adverse events (AEs) (i.e., nausea and vomiting) are more likely to attain greater weight reduction. Given that GI AEs occur more commonly in females than in males, the sex-related differences in GI AEs may offer an additional framework to explain the increased efficacy of these drugs among women, independently from the level of exposure ([76]; and references therein). Indeed, to date, there are no published data supporting the necessity for sex-specific dosage recommendations for GLP-1 RAs.

In this context, treatment adherence is an additional factor that must be taken into account. Males had lower treatment interruption rates and higher adherence rates than females, according to a retrospective study that looked at GLP-1 RA adherence and interruption rates [90]. These findings suggest that adherence may affect GLP-1 RAs’ effectiveness differently depending on sex, possibly preventing women from receiving the full therapeutic benefit. Consequently, additional fundamental differences in treatment outcomes may go unnoticed if a sex-specific pattern in adherence rates is present.

Finally, referring to sexual dimorphism in weight loss from another point of view, it has been highlighted that estrogen signaling plays a crucial role in the food-reward aspect of food seeking induced by GLP-1 RAs [91]. Accordingly, higher levels of estrogen, especially during premenopausal years, could probably explain why females may respond more effectively to GLP-1 RAs. Studies on humans suggest that food consumption tends to be lower during the follicular phase and the periovulatory period, but higher during the luteal phase, demonstrating an opposite correlation with estrogen levels [92]. These findings lend support to the hypothesis. At the moment, treatment with single-molecule peptides constituted by a combination of estrogen and GLP-1 RAs is under evaluation.

Hence, future research is warranted to further elucidate the impact of all the above factors (exposure, adherence, estrogen sensitivity of the brain to GLP-1) on the sexual dimorphism of treatment with GLP-1 RAs, and, concomitantly, to obtain a better understanding regarding the sex-specific mechanisms of the action of these drugs.

#### 6.2.3. Effects on Subcutaneous vs. Visceral Fat Depot

Mounting evidence indicates that GLP-1 RA treatment leads to a comparable absolute and higher-percentage decrease in VAT in comparison with SAT, implying that GLP-1 RAs may trigger other pathways that calorie restriction does not [93].

A systematic review of 10 trials provided proof that treatment with GLP-1 RAs can considerably lower VAT and SAT in subjects with T2D. Interestingly, treatment with GLP-1 RAs seemed to be associated with a prioritized decrease in VAT (11.2%) over SAT (8.3%) when baseline values for both VAT and SAT were considered. In addition, the subgroup experiencing significant weight loss saw a higher decrease in VAT and in SAT as a result of GLP-1 RA treatment. Furthermore, there was a noteworthy correlation between overall weight loss and the VAT reduction [93].

Still, the occurrence of a sexual dimorphism in body fat reduction induced by GLP-1 RA treatment remains an interesting but still a poorly explored issue.

When considered collectively, these results suggest that decreases in visceral and hepatic fat may be the mechanisms underlying the benefit against CVD that some GLP-1 RAs have shown in T2D patients. In light of the recent discovery that visceral and ectopic fat are important cardiovascular risk factors, future weight-loss studies ought to incorporate targeted, gold-standard imaging of liver fat and VAT, which are valuable and adjustable targets for the treatment of obesity.

### 6.3. Brown and Bright Adipose Tissue: Other Potential Targets of GLP-1?

Recent research characterized the incretin system in fat tissue, and one intriguing finding was that GLP-1R appears to play a significant role in energy metabolism by directly promoting brown AT (BAT) and mitochondrial bioenergetics remodeling [94]. Brown adipocytes differ from white adipocytes in that they are composed of many small lipid droplets, have a high concentration of mitochondria, and, most importantly, express high levels of uncoupling protein 1 (UCP1), which is essential for non-shivering thermogenesis. BAT uses circulating substrates like fatty acids, glucose and some amino acids in addition to TAG during this process. Therefore, BAT activation could potentially improve specific metabolic parameters (hyperglycemia, dyslipidemia) in addition to increasing energy expenditure, making it a viable therapeutic target for obesity and other metabolic diseases [95].

Another different form of adipose tissue can be highlighted: the beige or brite (“brown in white”) adipose tissue [96]. Two different processes lead to this intermediate form: the trans-differentiation of mature white adipocytes into brown adipocytes or the induction or differentiation of adipocyte progenitor cells. Considering the relatively modest quantity of activatable BAT in humans, the browning of white AT could represent a fundamental mechanism to augment thermogenic capacity [96].

Preclinical research showed that the main mechanism by which GLP-1 causes weight loss is not only by decreasing appetite and hunger but also by increasing energy expenditure, most likely by means of BAT thermogenesis [95].

Recent animal studies showed that the intraperitoneal administration of liraglutide increased BAT oxygen consumption through the up-regulation of UCP-1 protein levels. Liraglutide primarily increases the activity of BAT type 2 deiodinase, indicating that it may activate BAT by boosting intracellular thyroid activation and thereby promoting weight loss [97]. Moreover, another study indicated that GLP-1 RAs are positively implicated in thermogenesis through the transient up-regulation of IL-6 [98].

In humans, two studies performed by using treatment with liraglutide and exenatide [99,100] evaluated the resting energy expenditure (REE) and BAT fat fraction of the supraclavicular BAT depot using magnetic resonance imaging (MRI) or a cold-induced ^18^F-FDG-PET/CT scan. When liraglutide treatment was compared to a placebo for 26 weeks, there was no significant reduction in the supraclavicular fat fraction. Furthermore, neither the fat fraction nor the volume of the entire cohort was affected by exenatide treatment. There are several reasons which might explain such apparently disappointing results. First, the dose of liraglutide used (1.8 mg/day) was reasonably lower than that currently recommended (3.0 mg/day) in obesity treatment, and thus possibly insufficient for BAT activation. Moreover, the possible BAT activation somewhere, as well as the lack of the measurement of intracellular lipid combustion by MRI, may have also contributed to explaining the results of GLP-1 RAs in BAT.

Remarkably, lean body mass reduction was documented in both studies, with fat-free mass being a significant component of REE. Nevertheless, the results showed reduced or unaltered REE in both studies after treatment with GLP-1 Ras, as opposed to the first studies, where GLP-1 RAs led to a raised REE, mainly when adjusted for fat-free mass [101]. This dissonance might potentially be clarified by the fact that the first studies were carried out with the use of different methodologies and during dissimilar periods of therapy; nonetheless, it is commonly believed that REE decreases after weight reduction [102]. Therefore, even after accounting for lean body mass, a plausible explanation for the decline in REE following 4 weeks of liraglutide therapy could be that REE attenuates weight loss by adapting to improve metabolic efficiency in response to decreased food intake [99].

More research is required to investigate GLP-1 RAs’ effect on BAT activation in obese individuals, most likely using innovative non-invasive imaging methods on a wider scale.

## 7. GIP and GIP Receptors: Novel Actors in Adipose Tissue (Patho)Physiopathology?

Despite similarities to GLP-1, GIP has not been extensively evaluated as a potential modulator of adipose tissue (dys)function. However, the presence of GIP-R on fat cells was shown in cell culture, animal and human models by gene expression and functional assays [103,104,105]. Increased glucose uptake, lipoprotein lipase activity and lipogenesis (i.e., the re-esterification of free fatty acids (FFA) into TAG in vitro in isolated fat cells) are all facilitated by GIP [105,106,107]. Furthermore, GIP-R expression increases with the differentiation of adipocytes [107]. In contrast, other in vitro research highlighted that GIP also supports lipolysis [108]. While some research views GIP’s action as beneficial and suggests that it could aid in the growth of healthy SAT depots, other studies claim that it may support the expansion of healthy SAT depots [109], whereas other studies consider GIP as an endogenous hormone that promotes obesity in AT [110,111]. Research on mice indicated that obesity induced by a high-fat diet can be reduced by blocking a selective deletion of AT GIP-R; however, these studies also found a link between obesity and a high-fat diet, which could lead to independent effects [112]. Consequently, GIP increases the ectopic fat depot in mice, which is a sign of unhealthy obesity. It may also mediate the unfavorable consequences, not only of high-fat diets [112], but also of quickly assimilated or high-glycemic-index carbohydrates, a feature of highly processed foods that are commonly found in Western diets [51]. These findings undoubtedly show how endogenous GIP functions, but they might not provide useful information about a pharmacological agonist or antagonists [113].

In healthy lean humans, when infused alone, GIP did not modify the local metabolism in AT but it has been able to enhance the hydrolysis of TAG induced by insulin, glucose uptake and blood flow, at the same time reducing FFA output. TAG accumulation occurs during hyperinsulinaemic hyperglycemic clamps as a result of these metabolic processes [114]. Additional research verified that GIP did not alter energy metabolism or appetite, nor did it alter TAG clearance in the absence of insulin when compared with saline infusion of lipid emulsions [114]. Accordingly, other research highlighted the insulin-dependent effects of GIP on adipose tissue blood flow and metabolism, through insulin inhibition mediated by the administration of somatostatin [115].

The different effects of GIP in AT may therefore be associated with insulin levels, insulin sensitivity and BMI. As a matter of fact, obesity appears to generate resistance for the effects of GIP in AT: GIP-R density and GIP activity reduce in obese subjects, whereas they may increase after weight loss [116]. Ceperuelo-Malaffré et al. [117] proved that GIP-R expression is downregulated in SAT in obese individuals and negatively associated with BMI, waist circumference, systolic blood pressure, glucose and TAG values. In addition, the homeostasis model assessment of IR, glucose and G protein-coupled receptor kinase 2 (GRK2) appeared as variables that were robustly linked with GIP-R expression in SAT. According to immunoprecipitation experiments, GIP decreases GRK2’s binding to insulin receptor substrate 1 and increases GRK2’s interaction with GIP-R. Human fat cells cultured in hypoxia lacked the GIP actions reported under normoxia. In support of this, human adipose-derived stem cells from lean individuals showed increased insulin sensitivity as a result of GIP. Furthermore, GIP induced the expression of GIP-R, concomitant with a decrease in the activity of the incretin-degrading enzyme DPP4. These physiological actions of GIP were not identified in human fat cells derived from an obese milieu characterized by diminished amounts of GIP-R.

Another crucial topic regarding the role of GIP on AT is inflammation. Studies on cells confirm that GIP has pro-inflammatory properties. Indeed, adipocytes derived from human stem cells may secrete cytokines, such as IL-6, in response to GIP [108]. Co-cultures of human fat cells and human macrophages showed the augmented release of monocyte chemoattractant protein-1 (MCP-1), a chemokine implicated in the migration and infiltration of monocytes/macrophages, in the presence of GIP [116]. Furthermore, the overexpression of GIP-R in 3T3-L1 cells enhanced an increase in Jun N-terminal kinase (JNK), a mitogen-activated protein kinase (MAPK), and IR [118]. There is evidence that GIP-R is expressed in endothelial cells, monocytes/macrophages and stromal vascular AT. It has been suggested that these cells reciprocally interact to induce inflammation [119]. GIP-induced acute and restricted postprandial cytokines’ release may be viewed as a physiological response as part of the healthy role of AT [120]; when it is exaggerated, as in the case of insulin-resistant obesity, it becomes pathogenic [119]. On the other hand, the targeted deletion of GIP in myeloid immune cells resulted in increased inflammatory responses, suggesting that endogenous GIP may have an anti-inflammatory role [121].

The differences could be attributed to the different roles that GIP plays in the development of immune cells with respect to the myeloid compartment [122] and in an AT milieu where it promotes monocyte differentiation into more M1 or less M2 inflammatory macrophage phenotypes [116,119]. Notwithstanding, significant general GIP overexpression prevents obesity and lowers inflammatory markers in mice [123]. This illustrates the paradoxical similar effects of GIP agonists and antagonists on metabolism that remain to be resolved.

Yet, also in humans, evidence suggests that the pro-inflammatory activity of AT in obesity is tightly connected to IR [119]. In vivo studies on humans have demonstrated that the administration of GIP boosts the expression of chemokines like MCP-1 and cytokines like IL-6 [116,124]. Furthermore, increased levels of osteopontin (which is involved in T-cell activation and differentiation as well as macrophage activation) and a higher risk of CVD were linked to GIP-R polymorphisms and circulating levels of GIP [125]. It was postulated that GIP might promote endothelin-1’s (ET-1) release from endothelial cells, thereby increasing human osteopontin levels.

## 8. The Fine-Tuning of GLP-1/GLP1-R and GIP/GIP-R Axis in Fat: How to Reconcile This Challenging Puzzle?

As a corollary to these findings, it is possible to hypothesize that the “fine-tuning” of GLP-1R and the GIP-R signaling axis in AT may well represent an important framework in adaptive changes in energy metabolism and also in the regulation of glucose homeostasis.

On the one hand, as far as GLP-1/GLP1-R activation in AT appears important to preserve adipogenesis and fat mass at an “equilibrium” set point, the regulation of the GIP/GIP-R axis in fat tissue cooperates with GLP-1 in the modulation of systemic insulin sensitivity (Figure 1A).

In this setting, GLP-1 and its receptor may influence the development of adipocytes induced by nutrients, thereby affecting the whole-body energy metabolism. Nonetheless, the action of GLP-1R in directly stimulating mitochondrial bioenergetics and brown AT remodeling further supports the protective role of GLP1-R activation by promoting adipogenesis and increasing the capacity of adipose cells to store lipids, thus decreasing ectopic lipid overflow and enhancing IR.

On the other hand, during glucose absorption, GIP functions as an insulin-sensitizer incretin, operating in a mode comparable to that identified in lipid metabolism (Figure 2A).

Reduced sensitivity to GLP-1 and GIP in adipose precursor cells and in mature fat cells in obesity-associated insulin-resistant states indicates a “GLP-1/GIP-resistant” phenotype as a potential genetic predisposing factor for the development of clinical glucose intolerance and IR (Figure 1 and Figure 2B). Moreover, in the presence of insulin, GIP stimulates and engages the lipogenic machinery in “sensitive” adipocytes (Figure 1B), whereas the resistance to GIP/GIP-R signaling in fat cells leads to a breakdown of the lipogenic action, thus contributing to lipid spillover in remote organs with gluco-lipotoxic effects (Figure 2B). Finally, GIP may act as a “bifunctional “gluco-stat” to protect against glucolipotoxic injuries by promoting the main pancreatic hormones involved in glucose regulation and reducing episodes of high glucose concentrations, thus providing protection from these injuries [126].

## 9. Incretin-Based Therapy and Cancer Risk: Fact or Fake?

Preclinical studies revealed that because the thyroid gland expresses GLP-1R, GLP-1 RAs may have particular effects on it as well. Studies on carcinogenicity in rats and mice showed a dose- and time-dependent elevated risk of medullary carcinomas, primarily because of the increased release of calcitonin and the up-regulated expression of the calcitonin gene, which led to C-cell hyperplasia [127]. Based on these results, the U.S. Food and Drug Administration (but not the European Medicines Agency) contraindicated treatment with GLP-1 RAs in subjects who had a personal or family history of medullary thyroid cancer and multiple endocrine neoplasia type 2. But, the importance of these animal findings indicating the carcinogenicity of GLP-1 RAs has not been evidently proved in humans. The Liraglutide Effect and Action in Diabetes: Evaluation of Cardiovascular Outcome Results (LEADER) clinical trial and a meta-analysis of 12 clinical trials with liraglutide both reported a non-statistically significant increase in thyroid cancers [128,129]. Furthermore, two observational studies, as well as a study based on two U.S. administrative databases did not highlight a significant increased risk of thyroid cancer with exenatide [130,131].

A nested case–control analysis was conducted recently using the French national health care insurance system (SNDS) database. The analysis included all cases of thyroid cancer and subjects with T2D treated with second-line antidiabetic drugs between 2006 and 2018. The results showed that using GLP-1 RAs increased the risk of both medullary thyroid cancer and thyroid cancer overall, in particular after 1–3 years of treatment exposure (especially for male subjects) [132]. Also, the use of DPP4-i was found to be associated with a higher risk of thyroid cancer, leading the authors to hypothesize that this association might be partly explained by increased endogenous GLP-1 levels after the inhibition of DPP-4 [132]. Although this was the first study investigating the risk of thyroid cancer with GLP-1 RAs in a large administrative database, caution is required in generalizing these observations to clinical practice because of inherent study limitations (i.e., database misclassification and other potential confounders).

In contrast, recent meta-analyses of randomized control trials indicate the lack of signals related to the increase in the risk of pancreatic and breast cancer, in subjects treated with GLP-1 RAs either for T2D or for obesity [133]. Furthermore, a nationwide, retrospective cohort study including more than one million drug-naïve patients with T2D found that, during a 15-year follow up period, treatment with GLP-1 RAs was associated with a decreased risk of colorectal cancer compared with insulin, metformin, SGLT2-i, sulfonylureas and thiazolidinediones [134].

Overall, these findings may assist clinicians in addressing patients’ concerns regarding the possible but unproven relationship between GLP-1 RA therapy and the occurrence of cancer, as there is no clear indication of an elevated cancer risk with currently available GLP-1 RAs.

## 10. Concluding Remarks and Open Questions

Adipose organ plasticity is caused by complex mechanisms that are currently poorly understood. The existing literature offers robust scientific rationale for focusing on the novel role of the incretin axis in fat as a potential driver of metabolic complications associated with obesity.

The development of incretin-based medications induced different scientific efforts to clarify the mechanisms behind the therapeutic actions of GLP-1 RAs, yielding a large body of still-growing evidence. Nonetheless, based on emerging data, it is also time to rethink how the GIP system can be manipulated for therapeutic benefit. Such an interest also stems from the higher effect on glycemic control and weigh loss showed with GIP/GLP-1 co-agonists like tirzepatide [135], as well as the intriguing effects of retatrutide, the first GIP/GLP-1/GCGR (glucagon receptor) co-agonist [136]. However, different findings regarding the actions of GIP on AT are at odds with recent clinical results, and these divergences may be the repercussion of missing the tools used to evaluate the long-term effects of GIP-R stimulation [14]. Still, the reappraisal of GLP-1 and GIP pharmacology, and the possibility of crudely using adipose-specific drugs, normalizing the disruption of incretin signaling for inducing metabolic and cardiovascular improvements, is a therapeutic challenge that warrants further investigations.

## Figures and Tables

**Figure 1 ijms-25-08650-f001:**
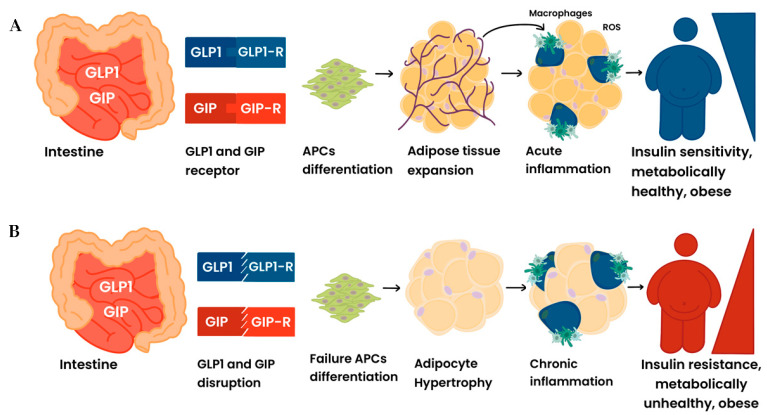
(**Panel A**): Following their intestine secretion, the activation of GLP-1/GLP1-R as well as that of GIP/GIP-R in adipose precursor cells appears important to preserve adipogenesis and fat mass expansion at an “equilibrium” set point. Healthy adipose tissue expansion induces a “metabolically healthy” obese phenotype. (**Panel B**): The disruption of GLP1/GLP1-R and that of GIP/GIP-R sensitivity in adipose precursor cells points to a “GLP1/GIP-resistant” adipose-specific phenotype as a predisposing factor to the development of metabolically unhealthy obesity.

**Figure 2 ijms-25-08650-f002:**
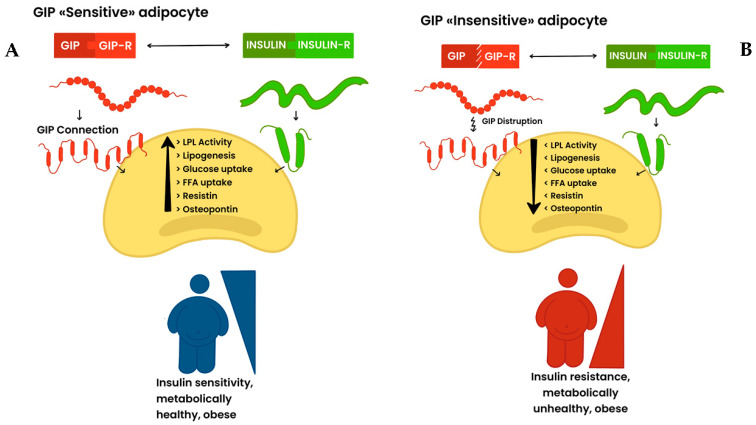
(**Panel A**): GIP «sensitive» adipocyte, GIP «sensitive» adipocyte. In mature fat cells GIP acts as an insulin-sensitizer incretin during glucose uptake by promoting and engaging, in the presence of insulin, the lipogenic machinery. (**Panel B**): GIP «insensitive» adipocyte. The disruption of GIP/GIP-R in fat cells leads to a breakdown of the lipogenic machinery, thus contributing to lipid overflow with gluco-lipotoxic effects.

**Table 1 ijms-25-08650-t001:** Tissue-specific effects of GLP-1 and GIP in animals and humans.

Effects of GLP-1	Tissues	Effects of GIP
↑↑ Insulin secretion ↓ Glucagon secretion	**Pancreas**	↑ ↑ Insulin secretion ↑ Glucagon secretion
↑ Glucose uptake, glycogen ↓ Hepatic glucose production ↓ Liver fat	**Liver (indirect effects)**	↑ Glucose uptake, glycogen
↑ Heart rate	**Heart**	↑ Heart rate
↑ Excretion of sodium	**Kidney**	No prominent direct effect
↓ ↓ Caloric intake Effects of anti-apoptosis and synaptic plasticity	**Brain**	↓ Caloric intake Effects of anti-apoptosis and synaptic plasticity (??)
↓ Gastric emptying	**Stomach**	No prominent effect
↓ Intestine motility	**Gut**	No prominent effect
↓ Inflammation	**Testicle**	No prominent effect
↑ Remodeling	**Bone**	↑ ↑ Remodeling

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
