# Peer review of "Adipose Tissue: A Novel Target of the Incretin Axis? A Paradigm Shift in Obesity-Linked Insulin Resistance"

_ijms, 2024, doi:10.3390/ijms25168650_

Round 1

Reviewer 1 Report (Previous Reviewer 2)

Comments and Suggestions for Authors

The authors have addressed all the points and the additional information have improved the quality of their work. I guess that the manuscript is now suitable for publication on International Journal of Molecular Sciences.

Comments on the Quality of English Language

English language has a good quality

Reviewer 2 Report (Previous Reviewer 1)

Comments and Suggestions for Authors

The revised manuscript has been significantly improved. This review article is summarized the mechanisms behind therapeutic actions of GLP-1 RAs role of incretin axis in fat as potential driver of obesity-linked metabolic. Great job.

This manuscript is a resubmission of an earlier submission. The following is a list of the peer review reports and author responses from that submission.

Round 1

Reviewer 1 Report

Comments and Suggestions for Authors

This  manuscript is well reviewed current studies of GLP-1/GIP receptor agonists in the treatment of insulin resistance/T2DM and its novel target in adipose tissues. 

Minors:

1. The figure 2A and 3A insulin sensitivity/resistance (i in healthy and unhealthy obesity cartoons are confuse. May redraw/modify them to make them clear.

2. May discuss sex differences in response to the treatment and its effects in subcutaneous vs visceral adipose tissues.

Reviewer 2 Report

Comments and Suggestions for Authors

The author has presented a clear review of the incretin hormones effects (GLP-1 and GIP) in the regulation of white and brown adipose tissue function. In addition, they discuss the scientific rationale for proposing adipose organ as a novel target for GLP-1 and GIP receptor agonists and co-agonists to prevent adiposity-linked metabolic complications. This manuscript is generally well-organized and attempts to answer an important question by reviewing the existing literature. However, some points should be further analyzed and better clarified:

11)     The authors should better discuss the molecular mechanism of GLP-1 and GIP action by adding a new paragraph. In addition, based on the major findings, authors could make a new figure to represent the molecular pathways of the incretin action.

22)    The authors have particularly focused their attention on adipose tissue. Some paragraphs need to be included in the manuscript about the role of GLP-1 and GIP in liver and muscle.

33)     As despite the metabolic benefits, an increased risk of cancer induced by incretin-based therapies has been reported. The authors should discuss this important point.

44)    It could be better to merge figure 2 with figure 3: in particular, panel 2A + panel 3A at the top and panel 2B + panel 3B at the bottom.

55)     As the authors indicate “several evidence” and include only one reference, we suggest adding more supporting references to some of the points like for example in lanes from 114 to 117, from 118 to 121 and from 176 to 178.

66)     The authors should better and deeply explain reasons why there is a greater metabolic disturbance associated with GLP-1R in morbidly obese and type 2 diabetic people but not in non-morbid obese or overweight subjects with insulin resistance. It could be also interesting to indicate the turnover point of this metabolic disturbance.   

Comments on the Quality of English Language

Quality of English Language is good